# Interference with *DGAT* Gene Inhibited TAG Accumulation and Lipid Droplet Synthesis in Bovine Preadipocytes

**DOI:** 10.3390/ani13132223

**Published:** 2023-07-06

**Authors:** Panpan Guo, Xuerui Yao, Xin Jin, Yongnan Xv, Junfang Zhang, Qiang Li, Changguo Yan, Xiangzi Li, Namhyung Kim

**Affiliations:** 1Guangdong Provincial Key Laboratory of Large Animal Models for Biomedicine, School of Biotechnology and Health Sciences, Wuyi University, Jiangmen 529020, China; gpanpan0307@163.com (P.G.); xueruiyao@yahoo.com (X.Y.); ynxu0613@163.com (Y.X.); 2International Healthcare Innovation Institute, Jiangmen 529020, China; 3Engineering Research Centre of North-East Cold Region Beef Cattle Science & Technology Innovation, Ministry of Education, Department of Animal Science, College of Agriculture, Yanbian University, Yanji 133002, China; jinxin1982@ybu.edu.cn (X.J.); junfangzhang0613@163.com (J.Z.); liqiang8589@ybu.edu.cn (Q.L.); 4Laboratory Animal Center, Yanbian University, Yanji 133002, China; 5Yanbian Hongchao Wisdom Animal Husbandry Co., Ltd., Yanji 133002, China; ycg@ybu.edu.cn

**Keywords:** tiacylglycerol, *DGAT*, preadipocyte, interference, transcriptome

## Abstract

**Simple Summary:**

Triacylglycerol (TAG) is the primary component of intramuscular fat, an important factor in determining meat quality. The synthesis of TAG is regulated by diacylglycerol o-acyltransferase (*DGAT*). This study investigated the regulatory mechanisms of two subtypes of *DGAT*, namely *DGAT1* and *DGAT2*, in the differentiation of Yanbian bovine preadipocytes and their roles in lipid metabolism-related signaling pathways. sh-DGAT1 and sh-DGAT2 were prepared using short interfering RNA (siRNA) interference technique targeting *DGAT1* and *DGAT2* genes and infected with bovine preadipocytes separately or simultaneously. Differentially expressed genes (DEGs) in bovine preadipocytes were analyzed using RNA sequencing and genome databases. During bovine preadipocytes differentiation, interference with *DGAT1* and *DGAT2* inhibited the formation of lipid droplets, content, and expression of lipid-forming genes at the mRNA level. *DGAT2* showed a stronger inhibitory effect. Transcriptome analysis revealed 2070 and 2242 DEGs between pre-adipocytes and normal cells that inhibited *DGAT1* and *DGAT2* expression, respectively, and 2446 DEGs when both were simultaneously inhibited. Our results indicate that compared with *DGAT1*, *DGAT2* plays a more important role in regulating bovine fat metabolism, which provides a theoretical basis for producing high-quality marbled beef.

**Abstract:**

Triacylglycerol (TGA) is the primary component of intramuscular fat. Expression of diacylglyceryl transferase (*DGAT*) determines the polyester differentiation ability of precursor adipocytes. The two *DGAT* isoforms (*DGAT1* and *DGAT2*) play different roles in TAG metabolism. This study investigates the roles of *DGAT1* and *DGAT2* in signaling pathways related to differentiation and lipid metabolism in Yanbian bovine preadipocytes. sh-DGAT1 (sh-1), sh-DGAT2 (sh-2), and sh-DGAT1 + sh-DGAT2 (sh-1 + 2) were prepared using short interfering RNA (siRNA) interference technique targeting *DGAT1* and *DGAT2* genes and infected bovine preadipocytes. Molecular and transcriptomic techniques, including differentially expressed genes (DEGs) and Kyoto Encyclopaedia of Genes and Genomes (KEGG) pathway analysis, were used to investigate the effects on the differentiation of Yanbian bovine preadipocytes. After interference with *DGAT1* and *DGAT2* genes, the contents of TAG and adiponectin were decreased. The TAG content in the sh-2 and sh-1 + 2 groups was significantly lower than that in the sh-NC group. RNA sequencing (RNA-seq) results showed 2070, 2242, and 2446 DEGs in the sh-1, sh-2, and sh-1 + 2 groups, respectively. The DEGs of the sh-2 group were mainly concentrated in the *PPAR*, *AMPK*, and *Wnt* signaling pathways associated with adipocyte proliferation and differentiation. These results demonstrated that at the mRNA level, *DGAT2* plays a more important role in lipid metabolism than *DGAT1.*

## 1. Introduction

The beef cattle industry plays an important role in China’s livestock industry and economy. Producing premium beef with rich intramuscular fat deposits, such as “Snowflake beef”, can enhance the profitability and prospects of the dairy-fattening business. The aim of any fattening operation is to produce meat with excellent quality characteristics, including tenderness, shear strength, and marbling [1]. These quality traits are associated with the intramuscular fat content of beef [2], indicating the importance of intramuscular fat content in determining meat quality. Intramuscular fat is mainly located in skeletal muscle fibers and stored as lipid droplets (LD) primarily composed of triglycerides (TAGs) and cholesterol esters [3].

TAG is the main form of energy storage in animals, and fat deposition in beef cattle muscle is mainly due to TAG accumulation. TAG is synthesized in vivo by two pathways: The glycerophosphate pathway (Kennedy pathway) in most cells and the glycerophosphoryl pathway in specialized cells [4]. These two pathways catalyze the synthesis of diacylglycerol and fatty acid acyl via the action of the microsomal enzyme diacylglycerol o-acyltransferase (*DGAT*) [5]. Studies have shown that *DGAT* is not only the last reactive enzyme controlling TAG synthesis in adipocytes but also the only key and rate-limiting enzyme in TAG synthesis [6]. *DGAT* comprises two subtypes, and the genes encoding these two enzymes are *DGAT1* and *DGAT2*. *DGAT1* belongs to the acyl-coA cholesterol acyl transferase (*ACAT*) gene family, while *DGAT2* to the acyl-coa monoacylglyceryl transferase (*MGAT1*) gene family, both of which have significantly different membrane topologies [7] and are widely expressed in various mammalian tissues where they play different roles [8]. Harris et al. [9] studied the effects of *DGAT1* and *DGAT2* genes on TAG synthesis and LD formation in mouse adipose cells and found that knocking out one gene in *DGAT1* and *DGAT2* does not interfere with TAG synthesis or LD formation in adipose cells. Conversely, when the two genes were simultaneously knocked out, no TAG was synthesized in the fat cells, and no LD were formed inside the cells. This study further demonstrates that *DGAT1* and *DGAT2* are closely associated with TAG synthesis, lipid deposition, and LD formation.

Given that enzymes in the *DGAT1* and *DGAT2* families have several different lipid acyltransferase activities [10], and since the functional activities of several of these enzymes are unknown, whether TAG is produced by enzymes other than *DGAT1* and *DGAT2* and the molecular mechanism by which *DGAT* regulates the differentiation and lipid metabolism of bovine preadipocytes remain unclear. We hypothesize that *DGAT2* plays an important role in TAG synthesis and storage and in regulating the differentiation of bovine preadipocytes.

In this study, sh-*DGAT1* (sh-1), sh-*DGAT2* (sh-2) and sh-*DGAT1*+ sh-*DGAT2* (sh-1 + 2) were generated via small interfering RNA (siRNA)interference targeting the *DGAT1* and *DGAT2* genes in infected bovine preadipose cells. Molecular and transcriptomic techniques were also used to investigate the role of *DGAT* in lipid metabolism and preadipocyte differentiation in cattle and to provide a theoretical basis for producing high-quality marbled beef.

## 2. Materials and Methods

### 2.1. Bovine Preadipocytes Isolation, Culture, and Differentiation

Preadipocytes were isolated from subcutaneous adipose tissue obtained from the backs of three 18-month-old Yanbian cattle using the collagenase digestion method [11]. Briefly, adipose tissue fragments of were sterilized with 75% alcohol and stored in a 15-mL centrifuge tube containing 1% phosphate-buffered saline (PBS; Gibco, Thermo Fisher Scientific, Waltham, MA, USA). Thereafter, the cells (5 × 10^4^) were inoculated in DMEM (Gibco, Thermo Fisher Scientific, Waltham, MA, USA) containing 10% fetal bovine serum (FBS; Gibco, Thermo Fisher Scientific, Waltham, MA, USA) and cultured at 37 °C under a 5% CO_2_ atmosphere. The culture medium was replaced every 48 h.

When the cell density reached more than 80%, lipid-induced differentiation was performed with a low serum concentration, and cell growth was observed and recorded on day 8 of culture.

In the induction differentiation process of different oleic acid concentrations, blank control group (CON, 5% FBS), 25 µM oleic acid treatment group (OA-25, 5% FBS + 25 µM OA), 50 µM oleic acid treatment group (OA-50, 5%FBS + 50 µM OA), 100 µM oleic acid treatment groups (OA-100, 5%FBS + 100 µM OA) and 200 µM oleic acid treatment groups (OA-200, 5%FBS + 200 µM OA) were repeated three times per treatment, and the differentiation medium was changed every 48 h. After 96 h of culture, transfected cells, and medium were collected for subsequent analysis to screen the optimal oleic acid treatment concentration.

At 80% confluency, cells were infected with either *DGAT1*-siRNA (sh-1), *DGAT2*-siRNA (sh-2), or *DGAT1*-siRNA + *DGAT2*-siRNA (sh-1+2) for 96 h. Preadipocyte differentiation was induced in a differentiation medium (DMEM supplemented with 5% FBS and 100 µM OA). After 96 h of culture, transfected cells, and culture medium were collected for subsequent analysis.

### 2.2. Construction of DGAT-siRNA

Based on the coding DNA sequence (CDS) regions of *DGAT1* and *DGAT2* in Yanbian cattle, three RNAi targets were selected. Primers for *DGAT1* were 144, 539, and 1157 bp, while those for *DGAT2* were designed and synthesized at 108, 320, and 818 bp. They were named siRNA-DGAT1-114, siRNA-DGAT1-539, siRNA-DGAT1-1157, siRNA-DGAT2-108, siRNA-DGAT2-320, siRNA-DGAT2-818, while shRNA-NC that did not target any gene was used as a control (siRNA-NC) and synthesized by Sangon Bioengineering (Shanghai) Co., Ltd. (Shanghai, China). Primer design was carried out through oligo software, and the primer sequences are listed in Table 1.

### 2.3. Detection of DGAT-siRNA Interference Effect

To determine a better interference effect, three DGAT1-siRNA and DGAT2-siRNA from the two genes were screened and evaluated. When the cell growth density reached >80%, cell infection was performed, and three replicates were used for each group. Cells were collected 48 h later, RNA was extracted, and infection efficiency was detected using fluorescence quantitative PCR to screen for the siRNA with the best interference effect for subsequent tests.

### 2.4. RNA Extraction and Quantitative Real-Time PCR (qRT-PCR) Detection

Total RNA was extracted from bovine preadipocytes using TRIzol™ reagent (Thermo Fisher Scientific, Waltham, MA, USA), and RNA integrity was resolved using NanonodropnD-100 spectrophotometer (2000C, Thermo Fisher Scientific, Waltham, MA, USA) and 1% agarose gel electrophoresis. A complementary DNA (cDNA) template was synthesized using the FastKing gDNA Dispelling RT SuperMix Kit (Tiangen Biotech, Beijing, China). The qRT-PCR was performed on an Agilent Mx3000/5p Real-Time PCR Detection System (Agilent Technologies, Santa Clara, CA, USA), using 20 μL SYBR Green SuperReal PreMix Plus (Tiangen Biotech, Beijing, China) containing 10 μL SYBR, 0.6 μL up- and downstream primers (10 μmol/L), 1 μL cDNA template, 7.5 μL Rnase-free ddH_2_O, and 0.3 μL Rox. PCR was performed as follows: Pre-denaturation at 95 °C for 15 min. There were 35 cycles of denaturation at 95 °C for 10 s, annealing at 55 °C for 30 s, and extension at 72 °C for 32 s. The dissolution stage was 95 °C for 15 s and 65 °C for 5 s. The relative expression of the target gene was normalized to that of glyceraldehyde-3-phosphate dehydrogenase (GAPDH; internal control) and calculated using the 2^−△△Ct^ method [12]. Primers for lipid metabolism- and phosphoglycerol pathway-related genes were designed online through primer premier 6.25 and synthesized by Shenggong Bioengineering Co., Ltd. (Shanghai, China), and their sequences are listed in Table 2.

### 2.5. Oil Red O Staining and Triglyceride Determination

An Oil Red O staining kit (G1262; Solarbio, Beijing, China) was used. Briefly, cells were washed with PBS, fixed with cytochrome fixative for 30 min, washed with 60% isopropanol, stained with Oil Red O staining solution for 10–15 min, and counterstained with Mayer hematoxylin solution for 1–2 min. Finally, cell images were collected under an inverted microscope (IX-73; Olympus, Tokyo, Japan) to observe the lipid droplets.

The triglyceride content was determined using an unkude triglyceride determination kit (Applygen Technologies, Beijing, China). The cells were washed three times with PBS and lysed in lysis buffer (R0010; Solarbio, Beijing, China). TAG content was determined via enzymatic colorimetry at 570 nm using a microplate reader (iMark; Bio-Rad, Hercules, CA, USA).

### 2.6. Determination of Aiponectin (ADP) Concentration

After 96 h of induction and differentiation, the cell culture medium was collected and analyzed using a bovine adiponectin ELISA Kit (Mlbio, Shanghai, China). A microplate reader (iMark; Bio-Rad, Hercules, CA, USA) was used to measure the optical density at 450 nm and generate a standard curve to determine adiponectin concentration.

### 2.7. RNA Sequencing (RNA-Seq)

Bovine preadipocytes were infected with sh-NC, sh-DGAT1, sh-DGAT2, and sh-DGAT1 + sh-DGAT2, with three replicates per group. Cell samples were collected 96 h after differentiation, and RNA was extracted using 1 mL of TRIzol reagent. Shanghai Parsonol Biotechnology Co. Ltd. (Shanghai, China) was used for RNA purification, cDNA library construction, and sequencing. Standard library preparation was sampled with an RIN value > 7.

The samples were sequenced by Illumina^®^ NovaSeq 6000 (2 × 150 bp ultra-long read length sequence has a better splicing effect), and the image file was obtained, which was converted by the software of the sequencing platform to generate the original FASTQ data, and each sample was analyzed separately. The sample name was Q30 with fuzzy base percentage, Q20 (%) and Q30 (%). There were some low-quality reads containing joints in the sequencing data, which might have interfered with the subsequent information analysis. Therefore, FastQC v0.11.8 was used to check the quality of the disembarkation data.

Reference Genome and Gene Model Annotation files (GTF files) were downloaded directly from the Genome website. Using HISAT2 v2.0.5 (http://ccb.jhu.edu/software/hisat2/index.shtml (accessed on 18 December 2021)) to construct the reference index in the genome, the paired-end clean reads were compared with the reference genome using HISAT2 v2.0.5. We used gene coverage parity and saturation analyses to assess the sequencing quality for the volume of the sequencing data. Under ideal conditions, the read distribution of all expressed genes should be uniform. We used RSeQC analysis of expression saturation to assess whether the amount of data measured was sufficient to correctly calculate gene expression levels. HTSeq (0.9.1) was used to calculate and compare the Read Count value of each gene with the original gene expression level. Non-stromalization of expression was performed using FPKM, and genes with FPKM >1 were generally considered to be expressed in the reference transcriptome. We used the DESeq software (version 1.20.0) to perform differential expression analysis between the two comparison combinations. For the gene expression via DESeq variance analysis, screening of differentially expressed gene conditions was as follows: Multiple expression differences |log2FoldChange| > 1, significant *p*-value < 0.05. The R language ggplots2 software package was used to map the volcano plots of the differentially expressed genes.

### 2.8. Gene Ontology (GO) and Kyoto Encyclopaedia of Genes and Genomes (KEGG) Enrichment Analysis

TopGO was used for GO enrichment analysis, and the hypergeometric distribution method was used to calculate the *p*-value (the standard of significant enrichment was *p*-value < 0.05) to determine the GO term of significant enrichment of differential genes so as to determine the main biological functions of differential genes. ClusterProfiler (3.4.4) software was used for the enrichment analysis of KEGG pathways, focusing on significantly enriched pathways with a *p*-value < 0.05.

### 2.9. Statistical Analyses

Data analysis and graph generation were performed using GraphPad Prism 6.07 (GraphPad Software, La Jolla, CA, USA) and SPSS Statistical software v19.0 (IBM, Armonk, NY, USA). The unpaired *t*-test was used for calculating *p* values. The results are presented as the mean ± standard error of the mean (SEM) from experiments performed in triplicate. Differences were considered statistically significant at *p* < 0.05.

## 3. Results

### 3.1. Interfering DGAT Gene Inhibited the Differentiation of Bovine Adipose Cells Induced by Oleic Acid

Prior to the experiment, we first cultured and identified the extracted preconditioned adipocytes in vitro. The results showed that the extracted precursor adipocytes had general cell growth characteristics (Appendix A), and the expression patterns of precursor adipocyte marker genes (Appendix A) and adipogenic marker genes (Appendix A) were consistent with the growth characteristics of precursor adipocytes, which could be used for subsequent experiments.

We performed sequential expression of *DGAT1* and *DGAT2* and found similar expression patterns between *DGAT1* and *PPARγ* and *C/EBPα*, that is, the expression levels of *DGAT1* gradually increased with the extension of differentiation time. However, the expression pattern of *DGAT2* was the opposite of that of *DGAT1* (Appendix A).

In the absence of oleic acid, only a small number of lipid droplets were formed in bovine precursor adipose cells on day 8 of differentiation (Appendix A). When 100 uL oleic acid was added, the accumulation of triglycerides and adiponectin significantly increased for 96 h (Appendix A), and the expression of adipose marker genes and related genes in the triglyceride synthesis pathway significantly increased (Appendix A). Therefore, the amount of oleic acid added was 100 uL.

The siRNA-mediated interference effects on *DGAT1* and *DGAT2* are shown in Figure 1A. The interference efficiency of siRNA-DGAT1-1157 with *DGAT1* was 77.5% compared with that of the negative control. Therefore, siRNA-DGAT1-1157 was selected for subsequent infection testing and was named sh-1. siRNA-DGAT2-108 of DGAT2 had the highest interference efficiency. Therefore, siRNA-DGAT2-108 was selected for subsequent infection tests and named sh-2. The experimental group coinfected with these two genes was named sh-1 + 2.

As shown in Figure 1B, different amounts and sizes of lipid droplets were formed in each experimental group due to the addition of oleic acid. The number and size of the lipid droplets in the sh-1 + 2 group were significantly lower than those in the other experimental groups. Meanwhile, compared with the control group, the contents of TAG and ADP in the sh-2 and sh-1 + 2 groups were significantly lower (*p* < 0.05) (Figure 1C).

The effects of *DGAT1* and *DGAT2* knockdown on the expression of genes related to the phosphoglycerol pathway synthesized by TAG are shown in Figure 1D. The expression of *DGAT1* in the sh-1 group was significantly lower than that in the other experimental groups (*p* < 0.05). However, there was no significant difference between the sh-2 and sh-1 + 2 groups (*p* > 0.05). The expression level of *DGAT2* in the sh-1 test group was the lowest. There was no significant difference between the sh-1 group and sh-2 group (*p* > 0.05), but there was between the sh-1 test group and the other test groups (*p* < 0.05). *AGPAT4* gene expression in the sh-NC and sh-2 groups was significantly lower than that in the other experimental groups (*p* < 0.05), and that in the sh-1 group was significantly higher than in the sh-2 and sh-1 + 2 groups (*p* < 0.05). *LIPIN1* gene expression in the sh-1 group was significantly lower than that in the other experimental groups (*p* < 0.05), but there was no significant difference between the sh-2 group and sh-NC groups (*p* > 0.05), while that in the sh-1 + 2 group was significantly higher than in the other experimental groups (*p* < 0.05). The expression of *MGAT1* and *GPAT4* genes in the sh-1 + 2 group was higher than that in the sh-NC group. However, the difference was not statistically significant (*p* > 0.05). The expression levels of *MGAT1* and *GPAT4* in the sh-1 and sh-2 experimental groups were the lowest. However, there were no significant differences between the two groups (*p* > 0.05).

The effects of *DGAT1* and *DGAT2* on the expression of lipid metabolism-related genes are shown in Figure 1E. The *PPARγ* gene expression in the sh-1 group was lower than that in the sh-2 and sh-1 + 2 groups, while the *C/EBP-α* gene expression was significantly higher than that in the sh-2 and sh-1 + 2 groups (*p* < 0.05). The *SCD* gene, sh-2 and sh-1 + 2 groups were significantly lower than the other experimental groups (*p* < 0.05). The expression level of *FABP4* in the sh-1 + 2 group was significantly higher than that in the other groups (*p* < 0.05), and the expression level in the sh-2 group was the lowest and significantly lower than that in the other groups (*p* < 0.05). The expression of *PLIN2* in the sh-1 group was significantly lower than that in the other experimental groups (*p* < 0.05), while the expression of *PLIN2* in the sh-2 and sh-1 + 2 groups was not significantly different (*p* > 0.05) but was significantly lower than that in the negative control group (*p* < 0.05). The expression of *CD36* in the sh-2 and sh-1 + 2 groups was significantly higher than that in the other groups (*p* < 0.05).

### 3.2. Difference Analysis of Bovine Preadipocytes Infected with sh-DGAT1/sh-DGAT2

Results from total RNA integrity tests showed that the quality of the extracted RNA was consistent with the requirements of the sequencing experiment for library construction (Figure 2A). Quality evaluation of the sequencing data showed that the proportion of high-quality clean reads was >91.00% in all the groups (Appendix A). The libraries were aligned against the Bos taurus (https://ftp.ensembl.org/pub/release-86/gtf/bos_taurus/Bos_taurus.UMD3.1.86.gtf.gz (accessed on 18 December 2021)) genome (Appendix A). The uniform-level gene coverage results showed no obvious bias toward the front (Figure 2B), and log10(FPKM + 1) showed a normal distribution (Figure 2C).

A total of 2070 DEGs were screened in the sh-1 group, including 1214 upregulated and 856 downregulated genes (Figure 3A). A total of 2242 DEGs were screened in the sh-2 group, of which 1255 were upregulated and 987 were downregulated (Figure 3B). In the sh-1 + 2 group, the total number of DEGs was 2446, of which 1317 were upregulated and 1129 were downregulated (Figure 3C).

### 3.3. Functional Analysis of Differentially Expressed Genes GO

The biological functions of DEGs were elucidated using GO enrichment analysis, and the 20 most significantly enriched GO terms were ranked according to the significance level. As shown in Figure 4, the biological processes (BP) of each treatment group accounted for most single gene annotations, followed by cell components (CC), and none of them were enriched in molecular function (MF). BP mainly includes multicellular organismal development and regulation of multicellular organismal processes, cellular developmental processes, cell differentiation, and cell surface receptor signaling pathways. CC enrichment mainly included the extracellular matrix, extracellular regions, and extracellular regions.

### 3.4. KEGG Enrichment Analysis of DEGs

KEGG pathway enrichment analysis showed that the main enrichment pathways of DEGs in the experimental groups included ECM-receptor interaction and the PI3K-Akt, MAPK, TGF-beta, and Hippo signaling pathways (Figure 5A–C).

### 3.5. Analysis of Signal Pathways Related to Differentially Expressed Genes and Lipid Metabolism

Transcriptome sequencing results showed that after *DGAT2* knockdown, several differentially enriched genes were associated with fat deposition in the selected pathways. After interference with *DGAT2*, *PPARγ*-regulated target genes also changed in the PPAR signaling pathway, and matrix metalloproteinase (*MMP1*), long-chain lipoyl CoA synthetase (*ACSL1*), peroxidase acyl-CoA oxidase 2 antibody (*ACOX2*) were downregulated. *FABP3*, *CD36* and oxidized low-density lipoprotein receptor 1 (*OLR1*) were upregulated (Figure 6A). There were 22 differential genes in the Wnt signaling pathway, among which 16 genes were upregulated, including *PLCB4*, *SFRP2*, *PRKCB*, *WNT2B*, *MYC*, *TCF7*, *DKK2*, *CCN4,* and *PLCB1*, and six were downregulated genes, namely *SFRP4*, *SFRP1*, *SERPINF1*, *WNT2*, *PRKCG,* and *PORCN* (Figure 6B). There were 12 differentially expressed genes in the AMPK signaling pathway, including seven upregulated genes and five downregulated genes. The upregulated genes were *PIK3CD*, *CREB3L3*, *FASN*, *CD36*, *HMGCR*, *STRADB,* and *PPP2R2C*, while those downregulated were *FBP1*, *INSR*, *PFKFB1*, *CFTR,* and *PFKFB3* (Figure 6C). Twelve DEGs were screened from the above signaling pathways for real-time PCR validation. The primer sequences are listed in Appendix A, while the results of real-time quantitative fluorescence PCR are shown in Figure 6A-C. The q-PCR results of different genes were consistent with the expression trends of the RNA-seq results.

## 4. Discussion

Short interfering RNA (siRNA) can bind to the target gene or messenger RNA of the target gene to inhibit expression with high specificity and efficiency [13]. TAGs are the most important form of energy storage in eukaryotic cells, most of which are stored in the LD of fat cells. As a key rate-limiting enzyme in TAG synthesis, *DGAT* plays an important role in lipid accumulation [14], and its expression level directly determines the polyester differentiation ability of fat precursor cells. *DGAT* includes two subtypes, *DGAT1* and *DGAT2*, which, although they catalyze the same biochemical reaction and have similar broad lipid acyl-CoA substrate specificity, play completely different roles in mammalian TAG metabolism [15]. In this study, small interfering RNA technology targeting *DGAT1* and *DGAT2* genes was used to infect bovine preadipocytes separately or jointly with sh-DGAT1 and sh-DGAT2, respectively, to explore their effects on TAG and lipid droplet synthesis during lipid differentiation. The interference effect of *DGAT1* was higher than 75%, and the interference effect of *DGAT2* was as high as 70%, which met the requirements for subsequent tests.

Unsaturated fatty acids and C-carotenoids from cattle feed can increase the expression of *PPARγ* and are natural activators of *PPARγ* [16]. As an agonist of *PPARγ*, the exogenous addition of oleic acid affects the expression of *DGAT*. However, the effects of oleic acid on the *DGAT1* and *DGAT2* genes of large mammal cattle remain nebulous. Different concentrations of oleic acid could promote the accumulation of lipid droplets in adipocytes of foie gras, as well as the expressions of lipid metabolism-related genes *DGAT1*, *DGAT2*, *PPARγ,* and *PLIN*, but with higher concentrations, the effect of oleic acid was gradually decreased [17]. In this study, the expressions of *DGAT1*, *DGAT2*, *PPARγ*, *CEBP/α*, *FABP4,* and *PLIN2* genes showed a significantly increasing trend with the increase of oleic acid concentration, and the higher the concentration, the greater the increase, which was contrary to previous studies [17], possibly due to the differences in species and cell sources.

The increased expression of *DGAT2* is associated with the formation of large lipid droplets, whereas the overexpression of *DGAT1* only produces small lipid droplets [18].

*DGAT2* has been shown to regulate the accumulation of TAGs in the tissues of DGAT1-deficient mice [19]. Additionally, *DGAT2* was highly expressed in various lipid-metabolizing tissues [20]. Moreover, *DGAT2* can compensate for LD formation in DGAT1-deficient intestinal stem cells [21]. In the present study, *DGAT2* overexpression increased the formation and accumulation of LD in Yanbian cattle preadipocytes, whereas *DGAT2* knockdown inhibited LD formation in the cells, which was consistent with previous findings [22]. Several studies have shown that low expression of *DGAT2* can cause a decrease in the TAG content of adipocytes [18].

*DGAT2* overexpression increased the expression of lipid-forming genes and the accumulation of TAGs in skeletal muscle cells (BSCs) [14]. In this study, inhibition of *DGAT2* expression was positively correlated with TAG and ADP content in preadipocytes. The importance of *DGAT2* in TAG synthesis was further explained [18].

Transcriptome sequencing is an important method for exploring gene function [23]. The *PPAR* signaling pathway plays an important role in regulating various lipid activities such as lipogenesis, fatty acid transport, and adipocyte differentiation. In yaks, there is a significant negative correlation between *FABP3* mRNA expression and MUFA level [24]. The transcriptome sequencing results of this study showed that the expression of *FABP3* in the pathway regulating lipid metabolism in the *PPAR* signaling pathway was significantly upregulated following interference with *DGAT2*. If the mRNA expression level of *FABP3* in Yanbian cattle is negatively correlated with MUFA levels, then the expression level of *FABP3* will be downregulated after the addition of oleic acid. However, the expression of this gene was upregulated after interference with *DGAT2*, which suggests a negative regulatory effect between *DGAT2* and *FABP3*. The regulation of the adipocyte differentiation pathway by *MMP-1* was significantly downregulated. We further speculate that the *DGAT2* gene plays an important role in lipid metabolism and differentiation regulation. The *ACSL* enzyme family is crucial to fatty acid metabolism in mammals and includes five members: *ACSL1*, *ACSL3*, *ACSL4*, *ACSL5,* and *ACSL6* [25]. *ACSL1*, involved in the activation of TAG fatty acid synthesis [26], is found in the liver, heart, and fat cells and has a wide range of fatty acid specificities [27]. In mice, overexpression of cardiac *ACSL1* increased the accumulation of TAG in cardiomyocytes by 12-fold [28]. Our sequencing results showed that following interference with *DGAT2*, *ACSL1* was significantly downregulated in the *PPAR* signaling pathway, consistent with the results of previous studies, indicating that *DGAT2* and *ACSL1* are positively regulated.

*AMPK*, an important cellular energy sensor [29], is a key factor in controlling cellular energy homeostasis and metabolism [30] and can reduce the expression of *SREBP-1*, *PPARγ*, and *C/EBp-α*, thereby inhibiting the accumulation of fat during fat formation [31]. *FASN* is a key enzyme involved in the process of fatty acid regeneration and plays a crucial role in energy homeostasis by converting excess carbon intake into fatty acid storage [32]. *CFTR* is a Cl- channel in the apical membrane of epithelial cells regulated by cAMP and protein kinase a (PKA). *AMPK* acts as a “biorheostat” of *CFTR*, that is, activation of *AMPK* can inhibit *CFPT* [33]. In addition, the loss of functional expression of *CFTR* is thought to upregulate *AMPK* activity in cystic fibrosis (CF)-deficient epithelial cells [33]. Transcriptional sequencing results showed that following interference with *DGAT2*, *FASN* expression in the *AMPK* signaling pathway was upregulated, while *CFPT* expression was downregulated. This is possibly because, after *DGAT2* expression is inhibited, intracellular energy is reduced, and *AMPK* activity is activated, which further promotes fatty acid synthesis, thus upregulating *FASN* expression and downregulating *CFPT* expression. This finding is consistent with those of the previous studies.

Wnt is involved in various important biological processes, including tissue regeneration, animal development, cell proliferation, and differentiation, and is a member of a conserved glycoprotein family. The typical Wnt/b-catenin signaling pathway is a highly conserved pathway critical to cell fate and patterns during development [34]. The catenin signaling pathway is regulated by a series of secreted molecules, including Wnt inhibitory signaling factor-1 (*WIF1*), Cerebrus, Sclerostin, Dickkopf-1 (*DKK1*) and secreted curl-associated proteins (*SFRP*). Sclerostin and DKK1 antagonize typical signals by binding to *LRP5/6*, whereas *WIF1*, cerebrus and *SFRP2* interact directly with Wnt proteins [35]. In mammals, *SFRPs* comprise five protein families: *Frzb (SFRP3)*, *SFRP1*, *SFRP2*, *SFRP4,* and *SFRP5* [36]. Since SFRPs are homologous to the WNt-binding domain sequence of the Fz receptor, they are considered typical Wnt signaling antagonists that can bind to Wnt proteins and block signal transduction [37]. However, it has recently been reported that *SFRPs* synergistically mimic Wnt activity by interacting directly with Fz receptors [38], antagonizing each other’s actions [39], enhancing the extracellular transport of Wnt proteins [40], or exerting other roles besides direct control of Wnt signaling pathways [41]. The sequencing results showed that *SFRP2* and *Wnt2* expression was upregulated following interference with *DGAT2*. Furthermore, the Wnt signaling pathway is closely related to the upstream gene process of lipogenic differentiation [42], which promotes the differentiation of mesenchymal stem cells into myoblasts or osteoblasts while inhibiting the differentiation of precursor adipocytes.

The identification of candidate genes is an important step in promoting marker breeding in Yanbian cattle. The results of this study confirm that *DGAT1* and *DGAT2* play important regulatory roles in adipocyte differentiation and lipid metabolism, providing insights into improving the bovine genome annotation and molecular breeding of beef cattle.

## 5. Conclusions

In this study, owing to the addition of OA, LD generation was observed after the *DGAT1* and *DGAT2* genes were disrupted, the contents of TAG and ADP were significantly reduced, and the expression of genes related to fat metabolism was inhibited at the mRNA level. RNA-seq was used to analyze the differentially expressed genes interfering with *DGAT1* and *DGAT2* in bovine preadipocytes, and 2070, 2242, and 2446 DEGs were detected in the sh-DGAT1 and sh-DGAT2 infected groups alone and co-infected groups, respectively. In the sh-DGAT2 treatment group, DEGs were enriched in the *AMPK*, *TGF-β,* and *PPAR* signaling pathways associated with adipocyte proliferation and differentiation, thus regulating the production of lipids by regulating the transcription of related genes in adipocytes.

## Figures and Tables

**Figure 1 animals-13-02223-f001:**
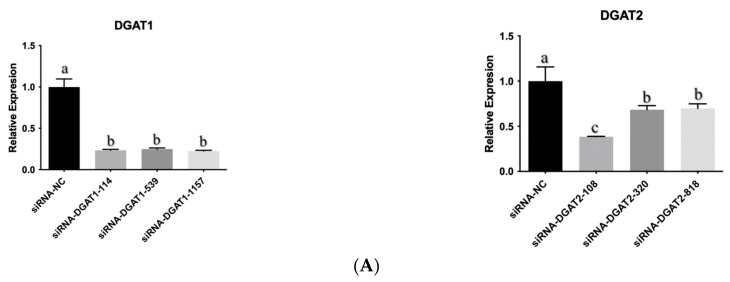
Effect of interference with *DGAT* gene on differentiation of bovine proadipocytes. (**A**) mRNA expression of *DGAT* post siRNA infection. (**B**) Oil red O staining (Scale bars: 200 µm). (**C**) Concentration of triglycerides and adiponectin. (**D**) Expression of genes associated with triglyceride synthesis pathway. (**E**) Expression of genes associated with lipid metabolism. Data are presented as the mean ± SEM (*n* = 3). The different letters (a–d) represent significant differences (*p* < 0.05) in gene expression.

**Figure 2 animals-13-02223-f002:**
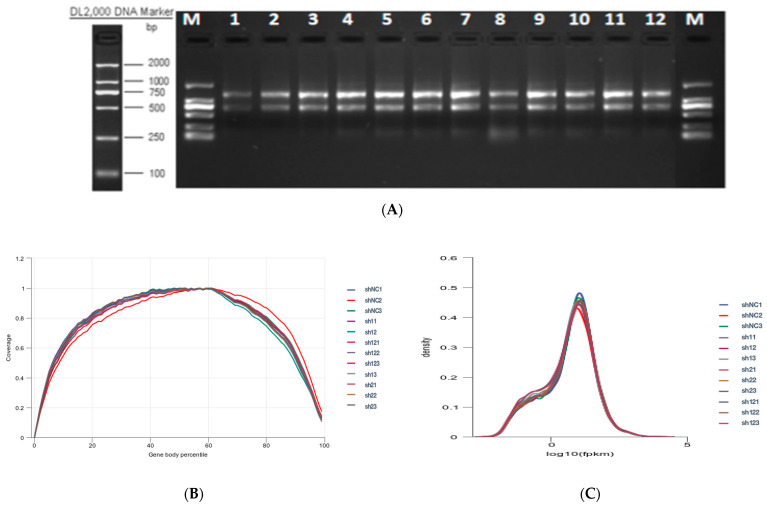
Total RNA quality detection and RNA-seq data quality assessment. (**A**) Total RNA integrity detection. (**B**) Gene coverage uniformity. (**C**) Density distribution of FPKM.

**Figure 3 animals-13-02223-f003:**
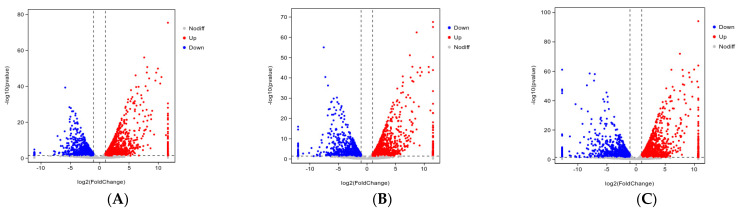
Volcano map of DEGs. (**A**) sh-NC vs. sh-1 groups, (**B**) sh-NC vs. sh-2 groups, and (**C**) sh-NC vs. sh-1 + 2 groups. Significantly upregulated DEGs are indicated with red dots, whereas downregulated DEGs are represented with blue dots. Gray dots indicate nonsignificant significant DEGs.

**Figure 4 animals-13-02223-f004:**
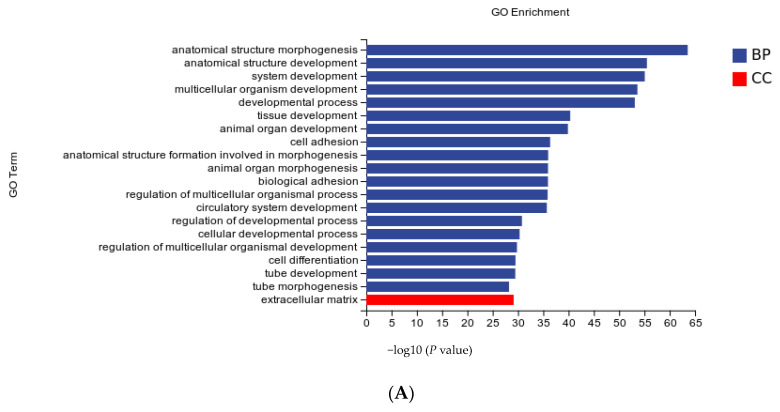
Gene ontology (GO) term analysis of the differentially expressed genes (DEGs). GO functional annotation of the DEGs in biological processes, cellular components, and molecular functions. The 20 significantly enriched GO terms. GO functional annotation of DEGs in the sh-NC vs. sh-1 groups (**A**), sh-NC vs. sh-2 groups (**B**) and sh-NC vs. sh-1 + 2 groups (**C**).

**Figure 5 animals-13-02223-f005:**
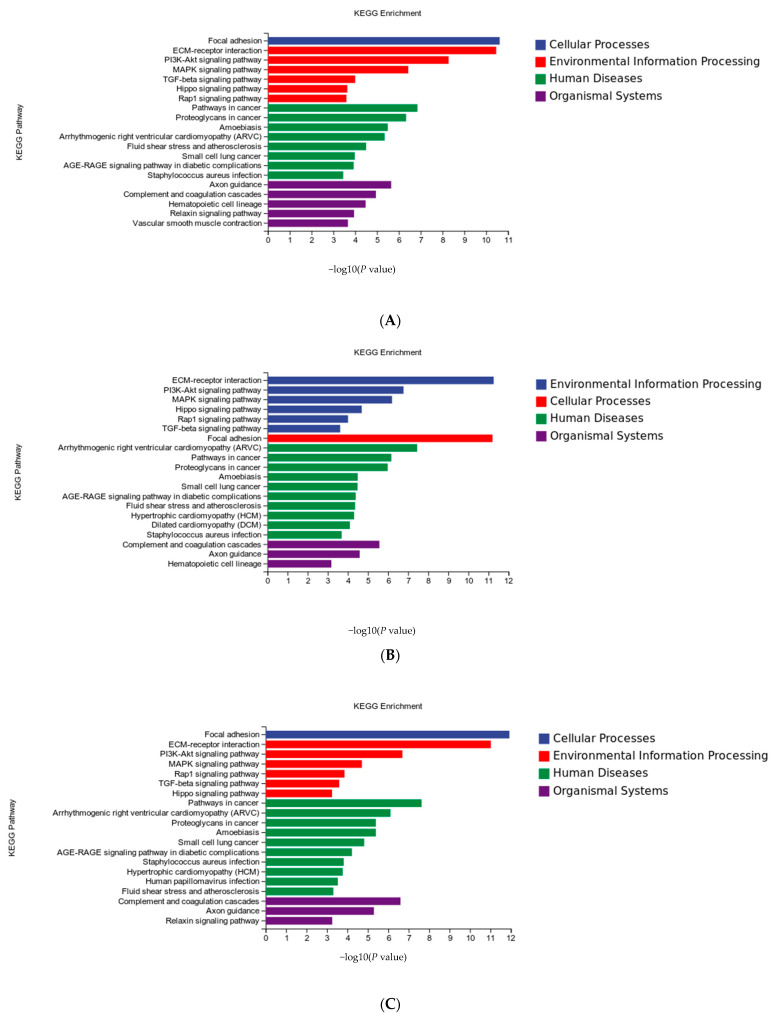
Kyoto Encyclopedia of Genes and Genomes (KEGG) pathway analysis of differentially expressed genes (DEGs) in the sh-NC vs. sh-1 groups (**A**), sh-NC vs. sh-2 groups (**B**) and sh-NC vs. sh-1 + 2 groups (**C**).

**Figure 6 animals-13-02223-f006:**
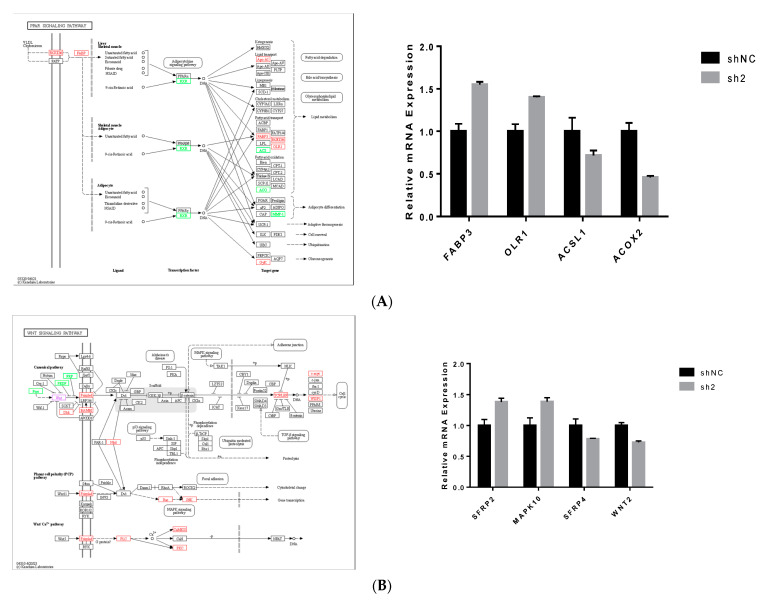
DEGs were enriched in PPAR, WNT, and AMPK signaling pathways (**A**–**C**). A total of 12 differential genes were screened and verified using real-time quantitative PCR (**A**–**C**).

**Table 1 animals-13-02223-t001:** siRNA sequences of *DGAT1* and *DGAT2* gene.

Scheme	Sequence	
	Sence (5′ to 3′)	Sence (3′ to 5′)
siRNA-DGAT1-144	AGACAAGGACGGAGACGUATT	UACGUCUCCGUCCUUGUCUTT
siRNA-DGAT1-539	CCUUUCUCCUCGAGUCUAUTT	AUAGACUCGAGGAGAAAGGTT
siRNA-DGAT1-1157	GCAUCAGACACUUCUACAATT	UUGUAGAAGUGUCUGAUGCTT
siRNA-DGAT2-108	GGUAGAGAAGCAGCUCCAATT	UUGGAGCUGCUUCUCUACCTT
siRNA-DGAT2-320	GCUACUUUCGAGACUACUUTT	AAGUAGUCUCGAAAGUSGCTT
siRNA-DGAT2-818	AGAAGAAGUUCCAGCUCCAATT	UACUUCUGGAACUUCUUCUTT
siRNA-NC	UUCUCCGAACGUGUCACGUTT	ACGUGACACGUUCGGAGAATT

**Table 2 animals-13-02223-t002:** Sequence information of PCR primers.

Gene	Sense Strand (5′→3′)	Length (bp)	Gene ID
*GAPDH*	F:ACTCTGGCAAAGTGGATGTTGTC R:GCATCACCCCACTTGATGTTG	143	NM_001034034
*DGAT1*	F:CTACACCATCCTCTTCCTCAAG R:AGTAGTAGAGATCGCGGTAGGTC	176	NM_174693.2
*DGAT2*	F:GACCCTCATAGCCTCCTACTCC R:GACCCATTGTAGCACCGAGATGAC	145	NM_205793.2
*AGPAT4*	F:TGTTCTCGTCTTCTTTGTGGCTTCC R:TCGCTATGTTTCTGCTTGCTGTCC	111	NM_001015537.1
*MGAT1*	F:AGCCGTGGTGGTAGAGGATGATC R:TGCTCCTTGCCATTGTCGTTCC	132	XM_024994376.1
*LIPIN1*	F:AGTCCTCGCCACACAAGATG R:AGATGCCCTGACCAGTGTTG	137	NM_001206156.2
*GPAT4*	F:ATGCGGTCCAGTTTGCCAATAGG R:GCTTCTGCTGCTCCTCCTTGAAC	129	NM_001083669.1
*PPARγ*	F:ATCTGCTGCAAGCCTTGGA R:TGGAGCAGCTTGGCAAAGA	138	NM_181024
*C/EBPα*	F:CCAGAAGAAGGTGGAGCAACTG R:TCGGGCAGCGTCTTGAAC	69	NM_176788
*PLIN2*	F:GCGTCTGCTGGCTGATTTCT R:TGTAAGCCGAGGAGACCAGA	139	NM_173980.2
*FABP4*	F:AAACTTAGATGAAGGTGCTCTGG R:CATAAACTCTGGTGGCAGTGA	134	NM_174314.2
*SCD*	F:TGCCCACCACAAGTTTTCAG R:GCCAACCCACGTGAGAGAAG	80	NM_173959
*CD36*	F:ACTGCGGATGGAATTTACAAAG R:ATGAGGCTGCATCTGTACCATTA	142	NM_001278621.1

## Data Availability

The datasets generated and/or analyzed during the conduct of the study are included in this published article. Upon reasonable request, the datasets of this study are available from the corresponding author.

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
