# Peer review of "Interference with DGAT Gene Inhibited TAG Accumulation and Lipid Droplet Synthesis in Bovine Preadipocytes"

_animals, 2023, doi:10.3390/ani13132223_

Round 1
Reviewer 1 Report
The manuscript entitled "Interference with DGAT gene inhibited TAG accumulation and lipid droplet synthesis in bovine preadipocytes" gives a very detailed insight into the mechanisms of fat accumulation in Yanbian cattle. The research appears to be a continuation of a study previously published in the journal Animals (Guo PP, Jin X, Zhang JF, Li Q, Yan CG, Li XZ. Overexpression of DGAT2 Regulates the Differentiation of Bovine Preadipocytes. Animals (Basel). 2023 Mar 29;13(7):1195. doi: 10.3390/ani13071195. PMID: 37048451; PMCID: PMC10093762). The article is well written, with an adequate introduction and well written materials and methods along with the results and discussion. Some minor suggestions need to be addressed before publication. You will find them in the attached file.

Author Response
Point 1: Please provide number of animals used for the study and number of samples generated for the analyses.
Response 1: Thank you for the keen review of our manuscript. We have added relevant information in the article.
Point 2: Please provide full name of the abbreviation.
Response 2: Thank you for the thorough review of our manuscript. We have added the full name of the abbreviation.
Point 3: please provide details on softwares used for the sinthesis of promers.
Response 3: Thanks for your comment. We have added information on primer design.
Point 4: in this sentence it was stated that primers were named siRNA1…, but in the table the names are different?
Response 4: Thank you for the thorough review of our manuscript. We have unified the names.
Point 5: please translate to english.
Response 5: Thanks for your comment. We have finished the English translation.
Point 6: Please provide details on how the primers were sinthesised (software etc)
Response 6: Thanks for your comment.We have added information on primer design.
Point 7: Please add more details on why these particular genes were chosen. This can only be observed from discussion, but I think it is important to add a bit more detail here.
Response 7:Thank you for the thorough review of our manuscript.We have made detailed modifications and have unified the full text.
Point 8-9: please split into two wordï¼›add refernces here.
Response 8-9: Thanks for your comment. We have revised the writing format and added the correct citations.

Reviewer 2 Report
TAG accumulation and lipid droplet synthesis are closely associated with meat quality and body health. This manuscript concerns the role of DGAT in lipid metabolism and preadipocyte differentiation in cattle. I am absolutely convinced that it is an interesting subject and falls within the scope of the journal.
However, before further consideration, the following question should be well addressed by the authors.
Is the section of “Simple Summary” necessary according to the guidelines of the journal? Please check.
I cannot find a clear, precise and solid conclusion at the end of the abstract.
In materials and methods, the purification method of the isolated cells should be clearly described. However, that is definitely important to directly determine the validity of the subsequent experimental results.
The data analysis strategy is dubious. Please state clearly which method is chosen to determine the statistical comparisons of the differences in TAG and ADP contents among treatments (Bonferroni correction, Tukey's multiple comparison test or other methods).
Table 2 should be reorganized to a standard three-line table. By the way, there may be a typesetting problem. All tables and figures in the manuscript should be rearranged to make the paper easy to read and understand.
The figures, especially figures 3, 4, 5 and 6, must be replaced with a high-resolution version. It is indistinct when enlarged the current pictures.
In contrast, the discussion part needs to further clarify the logic and go deeper.
Moreover, to help the reader better understand the underlying mechanism, the authors are encouraged to plot the diagrams of mechanisms involved in molecular dynamics.
Overall, the above issues led to the decision to reconsider it after major revisions that well addressed these concerns.
None.
Author Response
Point 1: Is the section of “Simple Summary” necessary according to the guidelines of the journal? Please check.
Response 1: Thanks for your comment. We have confirmed that “Simple Summary” is necessary.
Point 2: I cannot find a clear, precise and solid conclusion at the end of the abstract.
Response 2: Thank you very much for your comments. We have rephrased the conclusion section.
Point 3: In materials and methods, the purification method of the isolated cells should be clearly described. However, that is definitely important to directly determine the validity of the subsequent experimental results.
Response 3: Thank you for the keen review of our manuscript. The content of this part is supplemented in the form of supplementary experimental data in the result part.
Point 4: The data analysis strategy is dubious. Please state clearly which method is chosen to determine the statistical comparisons of the differences in TAG and ADP contents among treatments (Bonferroni correction, Tukey's multiple comparison test or other methods).
Response 4: Thank you for the thorough review of our manuscript. One-way ANOVA was used for determining the difference in TAG and ADP content among treatments.
Point 5: Table 2 should be reorganized to a standard three-line table. By the way, there may be a typesetting problem. All tables and figures in the manuscript should be rearranged to make the paper easy to read and understand.
Response 5: Thanks for your comment. We have rearranged all tables and figures.
Point 6: The figures, especially figures 3, 4, 5 and 6, must be replaced with a high-resolution version. It is indistinct when enlarged the current pictures.
Response 6: Thanks for your comment. All images have been replaced with the original images.
Point 7: In contrast, the discussion part needs to further clarify the logic and go deeper.
Response 7: Thank you for the thorough review of our manuscript. In the discussion part, we have reorganized and carried on the in-depth discussion.
Point 8: Moreover, to help the reader better understand the underlying mechanism, the authors are encouraged to plot the diagrams of mechanisms involved in molecular dynamics.
Response 8: Thank you for the keen review of our manuscript.We will continue to delve into the potential mechanism of DTAT regulating lipid metabolism and draw a schematic diagram of molecular dynamics mechanism for readers to learn from.

Round 2
Reviewer 2 Report
The subject of this paper is relatively popular and has the potential to become a highly cited article. However, for the consideration of the rigor of the paper, it seems that the description of the data analysis strategy in the paper is not very precise and clear, the layout of the figures needs to be further modified, and the discussion section of the paper needs to be further strengthened.
I strongly recommend that the author be patient and take a little more time and effort to address my concerns about the data analysis, the layout of the figures, and the logic and depth of the discussion, as doing so will help improve the readability and quality of the article.
None.
Author Response
Point:The subject of this paper is relatively popular and has the potential to become a highly cited article. However, for the consideration of the rigor of the paper, it seems that the description of the data analysis strategy in the paper is not very precise and clear, the layout of the figures needs to be further modified, and the discussion section of the paper needs to be further strengthened.
Response:Thank you again for the keen review of our manuscript. We have made detailed modifications to address the issues you raised regarding data analysis strategies, figures layout, and discussion sections. If there are any shortcomings, we look forward to receiving your valuable feedback.
